# Biomethane Production from the Mixture of Sugarcane Vinasse, Solid Waste and Spent Tea Waste: A Bayesian Approach for Hyperparameter Optimization for Gaussian Process Regression

Mansoor Alruqi [1] and Prabhakar Sharma [2,*]

1 Department of Mechanical Engineering, College of Engineering, Shaqra University, Al Riyadh 11911, Saudi Arabia
2 School of Engineering Science, Guru Nanak Dev DSEU, New Delhi 110089, India
* Correspondence: prabhakar.sharma@dseu.ac.in

**Abstract:** In this work, sugarcane vinasse combined with organic waste (food and wasted tea) was demonstrated to be an excellent source of biomethane synthesis from carbon-rich biowaste. The discarded tea trash might be successfully used to generate bioenergy. The uncertainties and costs associated with experimental testing were recommended to be decreased by the effective use of contemporary machine learning methods such as Gaussian process regression. The training hyperparameters are crucial in the construction of a robust ML-based model. To make the process autoregressive, the training hyperparameters were fine-tuned by employing the Bayesian approach. The value of $R^2$ was found to be greater during the model test phase by 0.72%, assisting in the avoidance of model overtraining. The mean squared error was 36.243 during the model training phase and 21.145 during the model testing phase. The mean absolute percentage error was found to be under 0.1%, which decreased to 0.085% throughout the model's testing phase. The research demonstrated that a combination of wasted tea trash, sugarcane vinasse and food waste may be a viable source for biomethane generation. The contemporary methodology of the Bayesian approach for hyperparameters tuning for Gaussian process regression is an efficient method of model prediction despite the low correlation across data columns. It is possible to enhance the sustainability paradigm in the direction of energy security via the efficient usage of food and agroforestry waste.

**Keywords:** machine learning; optimization; sustainability; energy production; circular economy



## 1. Introduction

Biofuels are greener fuels derived from bio-based resources and waste biomass which may be used to generate heat, power and fuel. With little modification, they may be utilized as transportation fuels [1]. Their abundance in nature, low cost, cleanliness and environmental friendliness have made biomass resources desirable [2,3]. Using thermal, biological and physical processes, biomass waste may be transformed into energy. Biogas is a common biofuel. Numerous nations give attention to the development of novel methods for producing biogas from biomass and biowaste [4,5]. Biogas created through waste biomass is believed to be the foundation of the future energy supply. Biogas can replace traditional fuels to provide heat and electricity. Biogas is used in several fields, including automotive applications, energy generation and the manufacturing of chemicals and materials. In addition, secondary waste, which is a by-product of this process, is a high-value crop fertilizer. The generation of biogas using anaerobic digestion (AD) has substantial benefits over other bioenergy production methods [6,7].

Food waste has detrimental environmental and socioeconomic effects. At the environmental level, when food waste is disposed of in landfills, a tiny amount of carbon in the trash is stored, but massively unutilized methane emissions enhance global warming prospects [8,9]. Furthermore, food waste disrupts the biogenic cycle by polluting the soil

and water with pathogens that develop in it. Food security, food waste-related costs and disdain for farmers' efforts are all key socioeconomic implications to consider. According to a World Commission on Environment and Development assessment, emissions and waste management issues might lead to an environmental disaster. Furthermore, governments have been firmly encouraged to articulate policies and coordinate measures via systematic attempts to alleviate these issues. Due to their large populations, insufficient food supply systems and ineffective management, China, the European Union (EU) and India are significant food waste-producing nations in the world [10–12].

Developing nations have more bioenergy production potential, allowing them to meet energy needs while also producing renewable and sustainable sources of clean energy fuels. Sustainable development, economic progress and environmental preservation are all important objectives that may be reached by reusing garbage. Thus, the bioenergy potential of agrifood waste contributes to the attainment of sustainable development goals (SDGs 7 and 13) since agriculture, food and their wastes may support the notion of a circular bioeconomy and net-zero waste by 2030. The most important waste solution is to convert agrifood leftovers into bioenergy. However, this requires a considerable initial investment in order to deploy appropriate and cost-effective approaches [13–15]. Tea waste (TW) is extensively employed in a variety of industries, including bioenergy generation, bioplastic manufacturing, polymer (mono and hybrids) composites, electrodes for supercapacitors and emission control applications. TW is a soil softener and can be used as a cultivation fertilizer that may also be used to remove contaminants from wastewater [16].

An alternative method of recycling used tea leaves is anaerobic digestion, which involves decomposing organic materials in the absence of oxygen. Biogas, the byproduct of this process, is a viable energy source that has several applications, including but not limited to thermal and electrical power generation. By combining them with other organic debris, used tea leaves may be composted to provide a nutrient-rich soil amendment. This may be used to boost plant growth and soil fertility [17,18]. The process of pelletization involves drying and compressing leftover tea leaves into pellets, which may then be utilized as a fuel source for both heating and generating energy. By burning used tea leaves in an incinerator, we can generate both heat and energy. When producing energy on a big scale, this method is often used. By heating used tea leaves in a low-oxygen atmosphere, a gas mixture comprising hydrogen and carbon monoxide may be produced. This process is called gasification. Power plants may utilize these gases as fuel. Biodiesel is made from spent tea leaves, which may be used as a renewable alternative to fossil fuels such as diesel. It is worth noting that some of these techniques may not be suitable for use in producing energy on a modest scale without more study and development [19,20]. The environmental effects of recycling used tea leaves should also be carefully considered. There has been little research in the domain of the co-composting/indigestion of TW. Tea farming in India spans around 5,79,000 acres, with an estimated annual tea yield of 857,000 metric tons, contributing to 27.51% of the total global tea production. As a result, the total TW generated from individual plants after the processing is approximately 190.4 k metric tons [13,19,21].

Composting is a bioconversion process that involves several different kinds of microorganisms that decompose waste sourced from organic feed and alter it into a stable product known as compost. The process of composting is an ecologically friendly way of managing organic solid waste. Compost provides a significant market for agricultural regions and polluted areas. The compost's stability and maturity are critical for its intended usage. An in-vessel composting approach is recommended because it reduces the requirements for space and extended time, which are the two most significant drawbacks. Various chemicals and earthworms may be used to improve the process efficiency and compost quality in this approach [22,23].

Though almost all organic wastes may be composted, owing to their physicochemical qualities, certain organic wastes can only be treated by co-composting. Co-composting has several benefits, including changing the content of initial moisture as well as the ratio of C to N, enhancing the efficiency of the process and enhancing the quality of the compost. As

a result, research in recent years has focused on the co-composting of multiple biogenic products together [24,25]. There has been a lot of study into the co-composting of various organic wastes but not a lot of research on the co-composting of tea wastes as well as food wastes. TW is a carbon-rich byproduct of the paper and pulp industry that is also called lignocellulosic biomass. TW co-composting is seen as a waste recycling procedure that is good for the environment. The final product could be employed as a soil amendment or fertilizer. So, it is important to look into several composting methods for recycling tea garbage [26,27].

Biomethane may be created from food waste and sugar cane vinasse via a process called anaerobic digestion. First, the food waste and sugar cane vinasse are mixed and put in a sealed container called a digester. The mixture is then heated to a temperature between 35 and 40 degrees Celsius, which produces an atmosphere that is conducive to the growth of anaerobic bacteria [28,29]. Methane and carbon dioxide are produced as a result of the breakdown of the organic materials in the mixture by the anaerobic bacteria. After this step, the methane is collected so that it may be converted into a source of renewable energy. The residual by-product, termed digestate, may be utilized as a fertilizer. It is worth mentioning that this procedure may be carried out on an industrial size or a smaller basis, such as on a farm utilizing a small-scale anaerobic digester [30,31].

Sugarcane vinasse is a byproduct of the sugarcane industry, and it can be used for biomethane generation through anaerobic digestion. This process involves microorganisms breaking down the organic matter in the vinasse in the absence of oxygen, producing methane and carbon dioxide as well as a nutrient-rich residue that can be used as a fertilizer. Biomethane can be used as a renewable source of energy for electricity and transportation. However, the quality of vinasse can vary depending on the sugarcane milling process, and it may require pre-treatment before being used for biomethane production [32,33]. Vinasse is the primary by-product during the ethanol manufacturing process, and it is created primarily during the step of distillation. Due to its significant organic content, strong potassium and sulfate levels, acidic and caustic characteristics and high pH, vinasse is both corrosive and alkaline. The latter occurs as a result of activities associated with the processing of the substrate for fermentation, such as the inclusion of sulfuric acid to control pH and prevent yeast flocculation [34,35].

Sugars that have not undergone a complete transformation retain a significant portion of their initial structure as well as their chemical makeup, which gives them a high fermentable potential. These sugars, such as glucose and fructose, are very readily accessible to enzymes and microbes, which enables them to break down and ferment the sugars into other molecules, such as ethanol and carbon dioxide. In addition, it is possible that these sugars have not been completely transformed by enzymes or other processes, which means that they are still in a condition that is readily able to be broken down and fermented [36,37]. Depending on whether the fermentable sugars used to make ethanol came from juice, molasses or a combination of the two, three different types of sugarcane vinasse may result from the industrial process. The rate of vinasse generation and fluctuations in the chemical oxygen demand (COD) of the biodegradable organic element found in vinasse are two indicators of this. These comprise organic acids, glycerol and sugars that were not completely transformed throughout the alcoholic fermentation, as well as any ethanol that was not recovered during distillation. The fermentation process leaves vinasse with a naturally acidic pH, which might hinder anaerobic digestion if not neutralized [5,38,39].

The co-digestion of biowaste/food waste for biomethane generation is a complex and nonlinear process. It depends upon several independent control factors. Hence, modeling through conventional methods is a difficult task. However, modern machine learning techniques can understand the pattern involved in the input–output paradigm of food waste/agrowaste [40–42]. Hence, in the present study, a modern ML technique is examined to model, predict and simulate the complex, nonlinear process of organic waste co-digestion.

## 2. Materials and Methods

### 2.1. Waste Collection and Preparation of the Substrate

The raw sugarcane vinasse was acquired from a rural-based sugar mill located near Delhi. Before use, the vinasse (raw) was refrigerated at $-5$ °C and analyzed following the usual procedure to estimate its approximate content and chemical oxygen demand (COD). The test substrate was watered down to a specific conc. of COD by mixing aqua dest beforehand and adding it to the chemical reactor, while the pH value was attuned to 7 by supplying about 50 mL of sodium hydroxide to every 1 L of the test substrate. Active methane-producing microbe inoculums are crucial for a quick and effective start-up of an anaerobic digester. The inoculum employed in this study was obtained from a working anaerobic digester based on cow manure, following a well-established procedure [36,43]. Before being supplied to the bioreactor, the inoculum was filtered to keep big particles out of the system. The conventional approach was used to characterize the inoculum to find the COD and volatile suspended solids. Solid tea waste (STW) is a potential waste from an organic source that is readily accessible in food factories, roadside tea shops, hotels, university campuses and practically all residences, particularly in India. STW is discarded in the form of wet slurry on open land after use. Its open dumping does not pose any great challenge or environmental hazard. However, in a circular economy, it is a waste that should be used for potential energy production. This was mixed with finely chopped-blended organic food waste collected from the hostel mess. The pigeon droppings were added to the slurry. Experiments were carried out in a conventional pilot-scale anaerobic digester. The pilot plant comprised a cylindrical digester with input and exit tanks and was constructed from FRP. The gas dome was constructed from three-layer reinforced fabric, consisting of high-tenacity rubberized nylon fabric covered with Hypalon on the outside and neoprene on the inside. Two segments of the FRP digester were manufactured and joined with a vertical junction on site. The junction was equipped with flanges with holes, and the segments were joined using bolts, with a 4 mm-thick rubber seal placed between the flanges to make the digester leak-proof. Table 1 depicts the properties of the feedstocks at various SV and STW proportions. The produced samples were stored in 5 L glass reactors with air-tight sealings. Figure 1 depicts the anaerobic digesters utilized in this investigation. In general, the trials were carried out at mesophilic temperatures. Before closing the digesters, all reactors were flushed with nitrogen for 10 min. A mechanical stirrer was employed to stir the mixture at regular intervals. Before measuring the biogas volume, each waste digester was physically stirred for 1 min twice a day.

**Table 1.** Feedstock's properties.

| Property | Sugarcane Vinasse | Spent Tea Waste | Food Waste |
|---|---|---|---|
| COD | 31.5, g/L | 21.05 | 32.5, g/L |
| pH | 4 | 6.7 | 5.45 |
| Total N | 0.56, g/L N | – | – |
| Ammoniacal N | 0.04, g/L | – | – |
| TOC | 20.14, g/m$^3$ | – | – |
| Volatile solid | 45.84, g/m$^3$ | – | – |
| C/N | 12.36 | – | – |
| TS, % | – | 5.65% | 13.14% |

COD: chemical oxygen demand; pH: power of hydrogen; C/N: carbon/nitrogen.

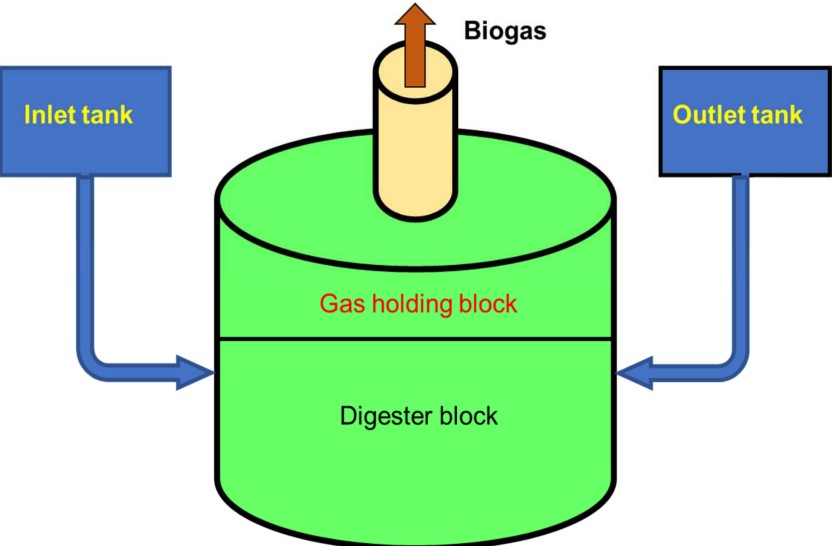

**Figure 1.** Biomethane digester.

*2.2. Soft Computing Method for Simulation Modeling*

2.2.1. Gaussian Process Regression

Carl Friedrich Gauss first developed the idea of a Gaussian process. The number of possible measurements for a Gaussian process is infinite, as is the case with other normal distributions. One method of machine learning that employs the kernel analysis of models is the Gaussian process [44,45]. In other words, it provides a workable method of studying kernel machines. During the fitting phase of a Gaussian Process Regressor, the kernel's hyperparameters are optimized by maximizing the log-marginal-likelihood using the past optimizer. There is a random variable set with a joint normal distribution for all finite components. The Gaussian process l has two main functions: the average function $m(x)$ and the function of kernel $n$ $(z, z')$ $(x)$. This approach incorporates the recompilation of the target variable in addition to the other attributes (where generalization is permitted) to facilitate the selection of a suitable noise level. Data gaps may be filled in using the global mean or mode. Binary characteristics are the translation of nominal qualities. If the kernel being utilized complies with Cached Kernel standards, kernel processing will be turned off [46–48]. A simple description of GPR is appended:

The GPR method is a form of a supervised non-parametric ML technique. The GPR is an outstanding technique based on the kernel system for training an implicit kind of data patterns for a large number of variables, making it especially well suited for handling challenging issues in nonlinear forecasting. The outcome '$O$' of the function '$f$' with the control factor '$x$' in the case of forecasting problems in GPR can be represented as [49–51]:

$$O_i = f(x_i) + \partial_i; \text{where } \partial \sim N\left(0, \varnothing_\partial{}^2\right) \tag{1}$$

The expression $f(x)$ is intended to represent the type of a random variable distributed depending on a certain GPR. Estimating the output of the function at many input positions can assist in reducing uncertainty regarding $f$. The terms are always influenced by a noise term that indicates their characteristic volatility. The data being assessed are shown in Equation (1) [52,53]:

$$Z = (x_i, y_i) \text{ and } i = 1, 2, 3, \dots, n.$$

Assume that $x_i$ and $y_i$ are scalar data, and $\partial_i$ is arbitrary and regularly distributed randomized inaccuracies having an average value of $\overline{\partial_i} = 0$ and a variance of $\partial^2$. Consider the fact that the measured $y_i{}'s$ values $(y_1, y_2, y_3, y_4, \dots, y_n)$ $T$ are predetermined type

values of the $f(.)$. As a result, $y_i's$ comprises a joint-type Gaussian distribution, as shown in Equation (2) [54,55]:

$$y = [y_1, y_2, y_3, y_4, \ldots, y_n]^T \sim N\left(p(x), K + \varnothing^2.I\right) \tag{2}$$

$p(x)$ represents the average of vector $p()$. $I$ is the unitary matrix, whereas $K$ signifies the covariance matrix ($\underline{n} \times n$) with the $(i,j)^{th}$ term $K_{ij} = k(x_i, y_i)$.

The ML technique GPR was implemented in the MATLAB program using suitable customization for this investigation. For tuning, a large variety of factors were explored, together with the kind of layers, the kernels and the activation function. The choice of the hyperparameters influences the efficacy of any ML method. In hyper-parameter tuning and grid or random search, evolutionary-type techniques are often used [56,57]. A significant number of function evaluations are required for these strategies. Figure 2 depicts the schematics of GPR.

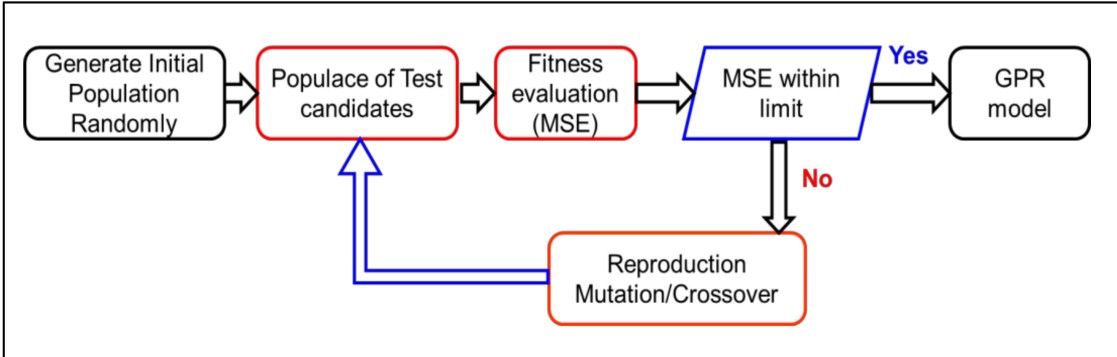

**Figure 2.** Schematics of GPR.

2.2.2. Bayesian Approach for the Optimization of Hyperparameters

A Bayesian optimization (BO) is a substitution approach to modeling and finetuning a complicated function for fitness by reducing the number of real function evaluations required. For closed-form equations in which the objective function is unknown but the data at sample values are known, the test method, which is grounded on Bayesian interpretation and Gaussian processes, may be of assistance [58,59]. BO seeks to identify optimal validation error-minimizing hyperparameters. BO uses constraints to determine the optimal value of a scalar goal function $f(x)$. The function can be stochastic or deterministic, meaning that it may produce a range of results when assessed around the identical point $x$. The variables making up $x$ might be string representations of names, actual values or unbounded integers [60,61].

The BO approach was employed in the present study for the optimization of the hyperparameters employed in the creation of prediction models. In the event of noise deviation, the sigma indicates the initial value for the constructed model. The MSE was utilized as a criterion for model selection, with $k = 5$. It can be employed to decrease a $'Y'$ such that $Y : X \rightarrow R$, while $X \in R^z$, where XRz. At a time-increment step '$t$' and location $x\ t$, the noisy function can be expressed. It may be written [62,63]:

$$y = f(x_t) + \partial_i \tag{3}$$

$\partial_i$ signifies the measurement noise in this context.

This technique employs a Bayesian-based prediction framework. In that way, the results of previous iterations are used to calculate the values for the following iteration.

Therefore, it might be more efficient than an arbitrary choice in developing the standard location. The principle can be phrased as follows:

$$p\left(\frac{t}{k}\right) = \frac{p\left(\frac{k}{t}\right)p(t)}{p(k)} \tag{4}$$

The preceding probability is denoted by *p(t)*, the proof by *p(k)*, the probability function by *p(k/t)* and the Bayesian probability by *p(t/k)*. Here, the function of acquisition—a parameter of the proxy model—is employed to select the next best place to assess. Those interested in finding a proven method may go to the current literature and find that Bayesian optimization is among them. Figure 3 depicts the BAO-based GPR flow chart.

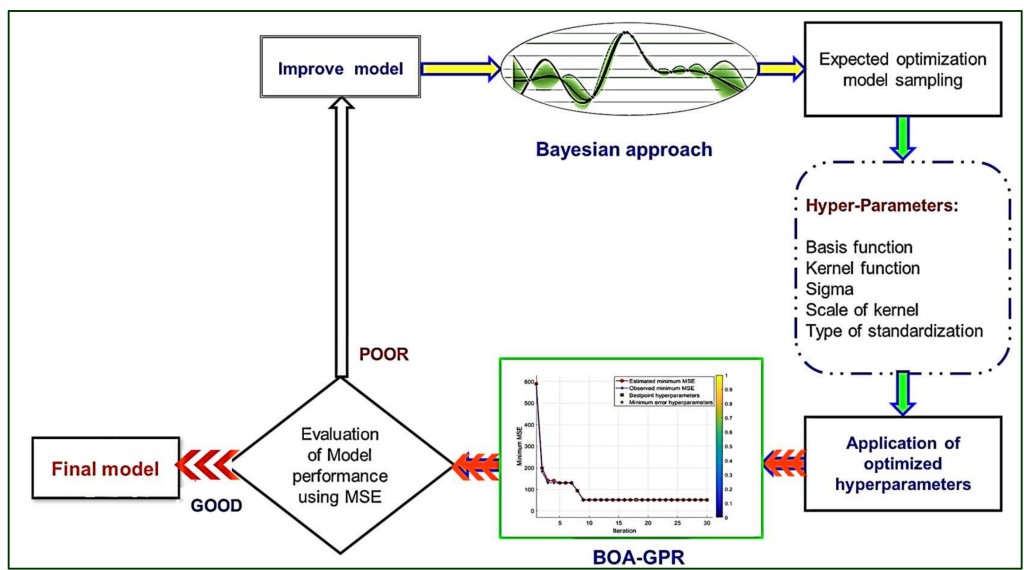

**Figure 3.** BOA-GPR flow chart.

2.2.3. Statistical Evaluation of the Prediction Model

A comprehensive set of statistical indicators was used in this study to explore the BOA-GPR's efficacy as a predictive tool. The coefficient of determination ($R^2$), mean absolute percentage error (MAPE), mean squared error (MSE) and root mean square error were used to estimate the efficacy of the forecasts generated by the recommended prognostic models. The values of $R^2$ should be as near to one as possible [64,65]. It is recommended that the MAPE be lower than 10%. The following phrase was used to arrive at these values for the indices.

$$R^2 = 1 - \left(\frac{\sum_{i=1}^{n}(x_o - x_p)^2}{\sum_{i=1}^{n}(x_o - x_m)^2}\right) \tag{5}$$

$$\text{MAPE} = \frac{1}{n}\sum_{i=1}^{n}\left|\frac{x_o - x_p}{x_a}\right| \times 100 \tag{6}$$

$$\text{MSE} = \sum_{i=1}^{n}\frac{(x_o - x_p)^2}{n} \tag{7}$$

$$\text{RMSE} = \sqrt{\sum_{i=1}^{n}\frac{(x_o - x_p)^2}{n}} \tag{8}$$

Herein, the total terms are $'n'$, $'i'$ denotes the term under consideration, $'x_o'$ represents the observed values, the predicted values are shown with $'x_p'$ and $'x_m'$ is the mean of the observed values.

## 3. Results and Discussion

### 3.1. Data Analysis

3.1.1. Kendall's Rank of Correlation

Kendall's correlation is a non-parametric technique for determining the connection between ranking data columns. Tau's coefficient yields 1 for the highest correlation between two columns and 0 for the lowest correlation. Kendall's rank correlation gives a measure of independence and a gauge of the intensity of the dependency between two variables that is not dependent on the distribution. Even if Spearman's rank correlation is sufficient for evaluating the null hypothesis of isolation between two factors, it is not easy to understand when the null hypothesis is rejected. This is because the null hypothesis was assumed to be true. Kendall's rank correlation adds to this by indicating the level of dependence that exists between the variables that are being considered [66]. In the present study, the parameters with a high correlation are depicted with red fonts. It was observed that a strong correlation exists between the percentage of vinasse as well as food waste % and biomethane production, as shown in Figure 4. The contribution of both vinasses, as well as food waste, is equally ranked, followed by the organic loading rate (OLR) and hydraulic retention time (HRT).

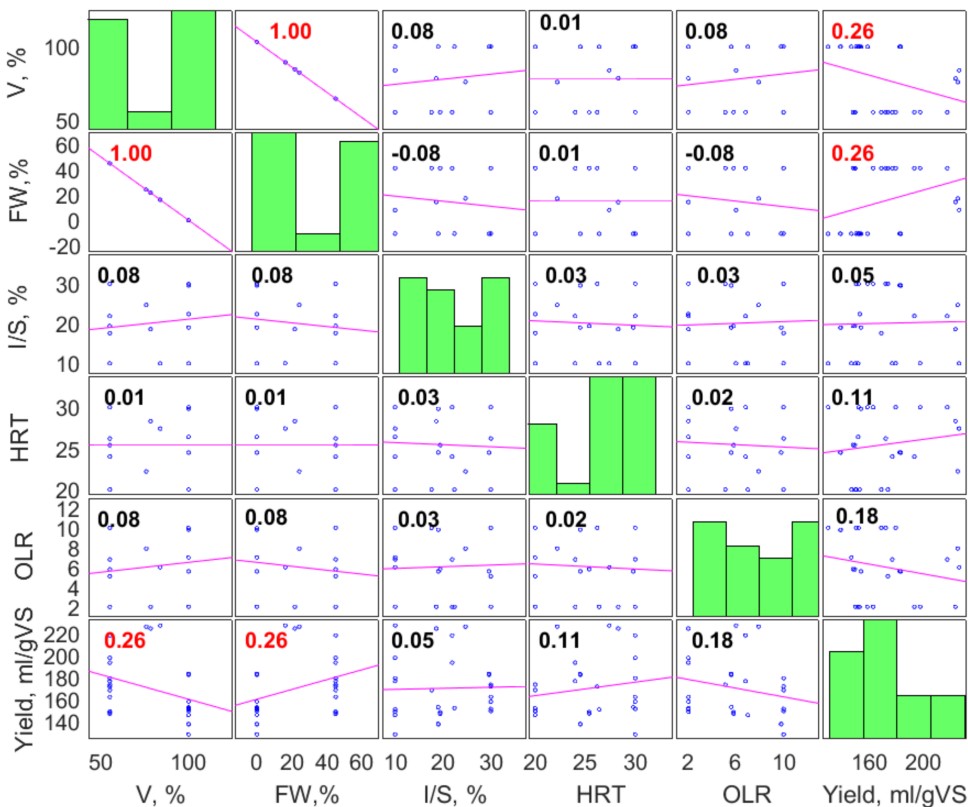

**Figure 4.** Kendall's rank of correlation.

3.1.2. Pearson's Correlation Matrix

Pearson's technique, in contrast to Kendall's, employs a linear method of correlation to demonstrate linear correlation from the perspective of parametric analysis. It shows the direct effect of parameters on the correlations among columns in a time series. The correlation heatmap for R is shown in Figure 5. Figure 5 complements the results of Kendall's rank analysis, as observed by the R values of vinasse and food waste, which were found to be numerically the same.

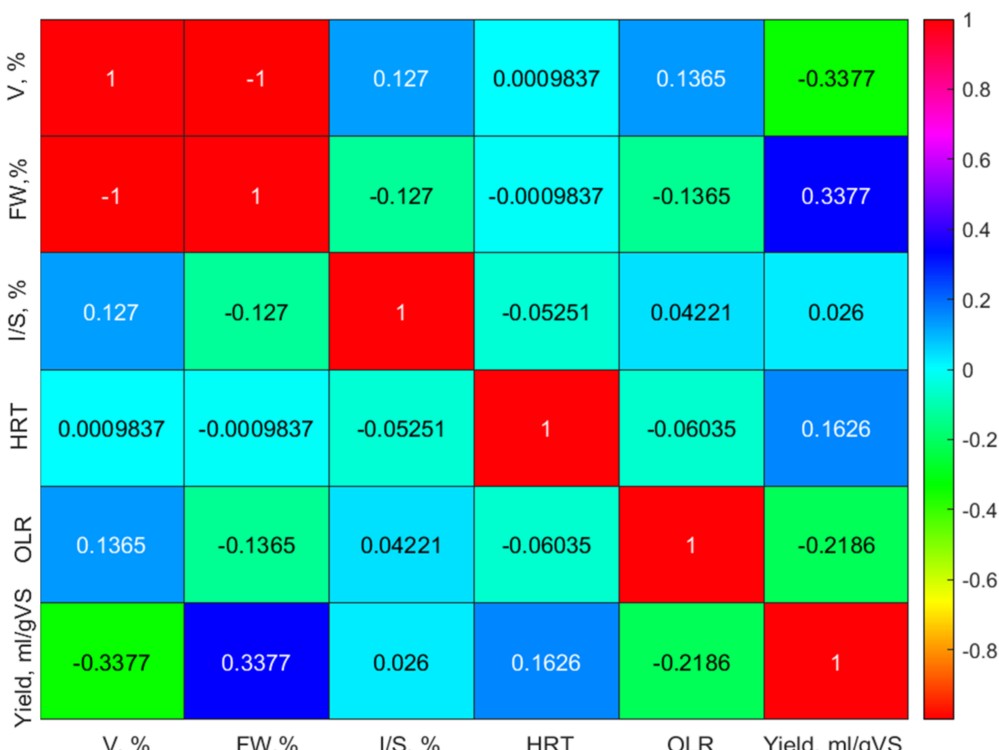

**Figure 5.** Correlation heatmap.

### 3.2. Model Prediction of Biomethane Yield

The ML technique GPR was used for the model prediction in this study. The training hyperparameters were optimized using the Bayesian approach for improved results. The five-fold cross-validation was used to prevent model overtraining and generalization. An autoregressive approach using the Bayesian approach was employed, and the model was trained using the least MSE as an optimization criterion. The training progress is shown in Figure 6. The details of hyperparameters and their finetuned settings are listed in Table 2.

**Table 2.** Training hyperparameters' environment and optimized settings.

| Hyperparameter | Hyperparameter Range | Set of Optimized Hyperparameters |
| --- | --- | --- |
| Preset | Bayesian approach | – |
| Number of iterations | 30 | – |
| Acquiring function | Potential improvements per second | |
| Time limit for training | False | |
| Functional basis function | Constant, Zero and Linear | Zero |
| Signal standard deviation | 19.93 | |
| Sigma value | 0.0001–281.76 | 0.00013 |
| Kernel function | Non-isotropic: Quadratic Rational, Matern 3/2 and 5/2, Squared Exp<br>Isotropic: Quadratic Rational Squared Exp, Matern 3/2 and 5/2. | Isotropic Matern 3/2 |
| Scale of Kernel | 0.045–45 | 0.9792 |
| Standardization | True, False | False |

The created model was validated by applying it to the experimental data to assess how well it performed. Figure 7 presents the results of the BOA-GPR-based model's performance. Figure 7 demonstrates that the model had satisfactory results. A reliable model may be inferred from the fact that the majority of the data points lie on the line that provides the greatest fit. Table 3 presents the results of the statistical analysis performed on the model. According to the numbers that are shown in the table, the statistical values will

improve when the model is put through its testing phase as opposed to while it is being trained. During the model test, the values of the correlation coefficient improved, whereas inaccuracies in the forecasting model decreased, suggesting that the model is capable of robust generalization.

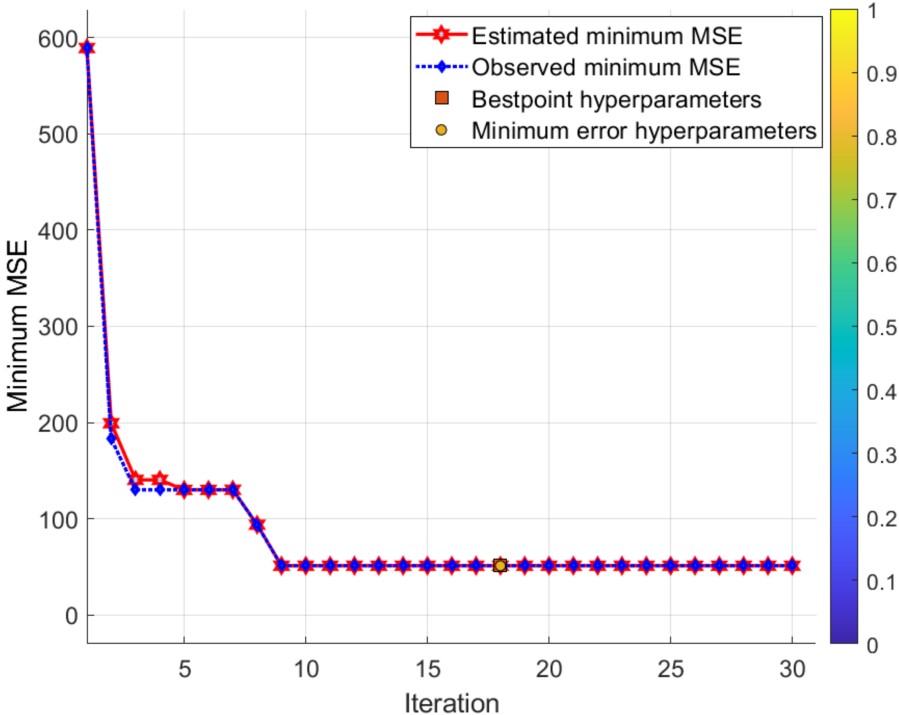

**Figure 6.** Model training progress.

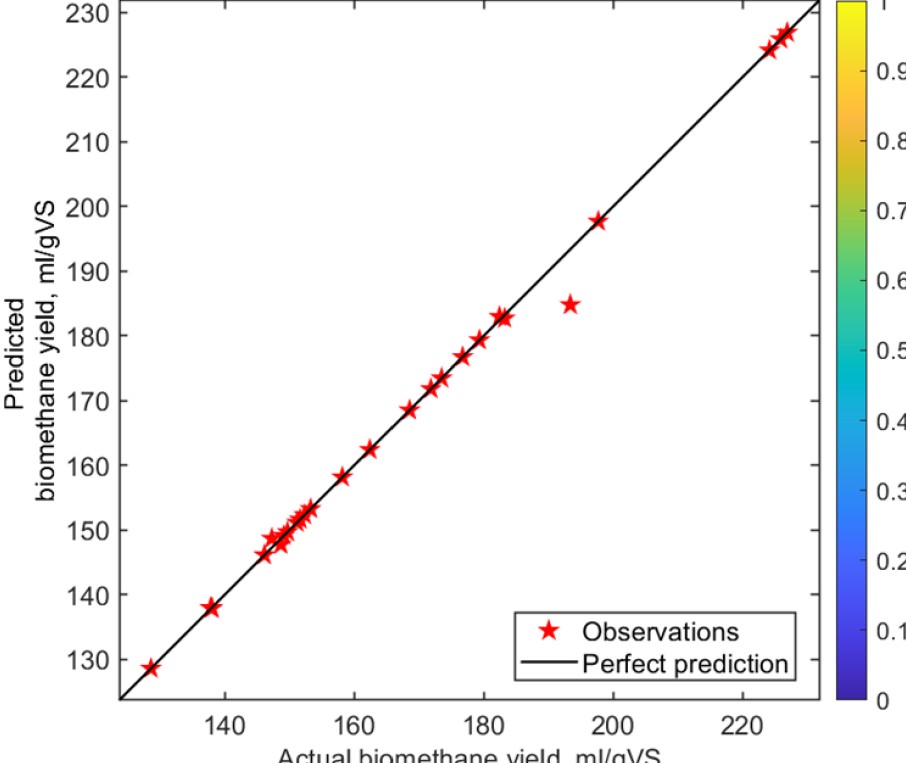

**Figure 7.** Model's performance.

**Table 3.** Model's efficiency and errors.

| Statistical Indices | Validation | Test |
|---|---|---|
| $R^2$ | 0.9478 | 0.9547 |
| MSE | 36.243 | 21.145 |
| RMSE | 6.0202 | 4.598 |
| MAPE | 0.1% | 0.085% |

The values of errors in the prediction data over the entire range of operations are shown in Figure 8. It can be observed that, except for a few points, the errors were quite low. The higher values of correlations and the low errors indicate that the BOA-GPR-based model is a robust prognostic model.

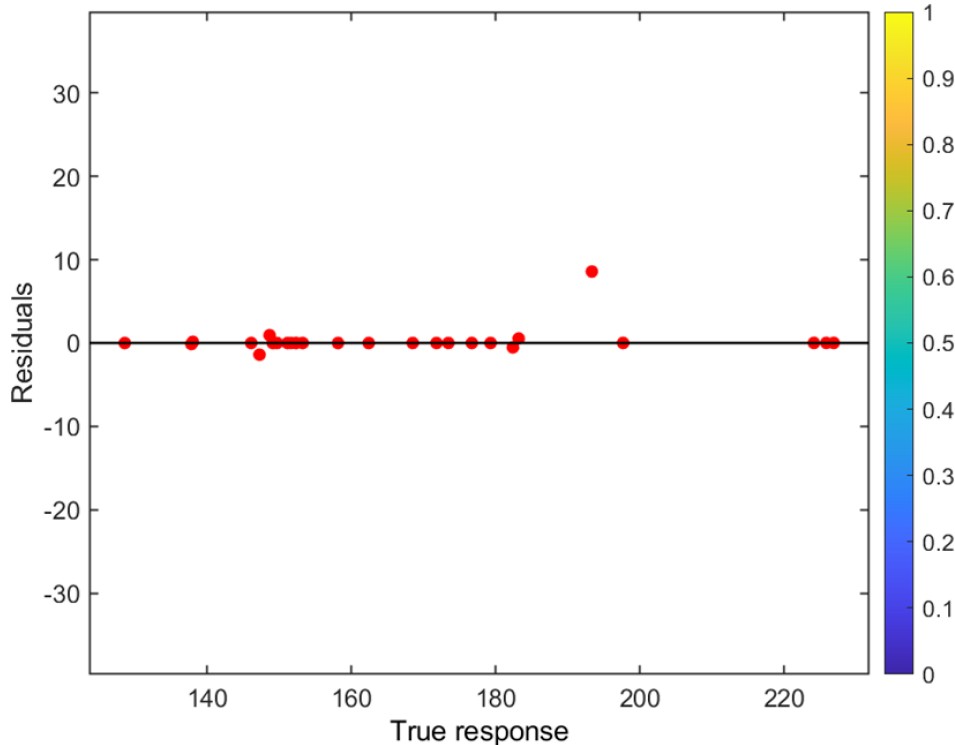

**Figure 8.** Errors in prediction values.

## 4. Conclusions

The sugarcane vinasse mixed with organic waste (food and spent tea) was employed in the present study for biomethane synthesis from carbon-rich biowaste. The uncertainties reduction and cost in the experimental testing were proposed to be reduced by the effective implementation of modern machine learning techniques, i.e., Gaussian process regression. The training hyperparameters play a significant role in the development of a robust ML-based model. The training hyperparameters were finetuned with the Bayesian approach to make the process autoregressive. The following are the main outcomes of the study:

○ Fivefold cross-validation helped in the prevention of model overtraining, as the value of $R^2$ was observed to be higher during the model test phase by 0.72%.

○ The mean squared error during the model training phase was 36.243, which was reduced to 21.145 during the model test process.

○ The mean absolute percentage error was observed to be only 0.1% which was reduced to 0.085% during the test phase of the model.

The study showed that the modern technique of the Bayesian approach for hyperparameters tuning for Gaussian process regression is an effective method of model prediction despite the poor correlation among data columns.

**Author Contributions:** Both M.A. and P.S. equally contributed to this study. All authors have read and agreed to the published version of the manuscript.

**Funding:** This research received no external funding.

**Institutional Review Board Statement:** Not applicable.

**Informed Consent Statement:** Not applicable.

**Data Availability Statement:** Not applicable.

**Acknowledgments:** The author (Mansoor Alruqi) would like to thank the Deanship of Scientific Research at Shaqra University for supporting this work.

**Conflicts of Interest:** The authors declare no conflict of interest.

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
