# Peer review of "Biomethane Production from the Mixture of Sugarcane Vinasse, Solid Waste and Spent Tea Waste: A Bayesian Approach for Hyperparameter Optimization for Gaussian Process Regression"

_fermentation, doi:10.3390/fermentation9020120_

Round 1

Reviewer 1 Report

Aunque el documento aborda un tema interesante mejora importante debe hacerse para aumentar su calidad.

La información entre las líneas 34 y 43 no se centra en el artículo Asunto. La información es demasiado general. Por favor, considere eliminar esas líneas.

La sección de metodología necesita correcciones importantes, la información es muy confuso; Por otro lado, el revisor pudo encontrar las variables utilizadas en el modelo, y hay frases que hacen muy difícil entender el procedimiento utilizado en los experimentos. Como ejemplo, consulte la página 4 línea 148-149.

Los autores deben ser más específicos sobre las características del reactor, señalando que "los experimentos se llevaron a cabo utilizando un anaeróbico convencional a escala piloto. digestor" no es suficiente. El mismo enfoque debería utilizarse con el análisis utilizado a través de los experimentos (por ejemplo, véase la página 3, líneas 133-134).

El significado de algunas abreviaturas debe mencionarse, por ejemplo STW, OLR, y HRT.

Hay muchos errores de mecanografía, por ejemplo: página 2, línea 83; página 3, línea 135; página 4 línea 163-164; página 5, línea 176; página 6 línea 237; página 8 línea 308; El documento completo debe ser revisados y los errores corregidos.

Las conclusiones necesita más consideración, faltan partes para apoyar algunas conclusiones; Por ejemplo, página 12 línea 356-358.

Sección de referencia necesita ser reconsiderado

Reviewer 2 Report

Dear authors, interesting work, I have one major concern and some minors. From my point of view, the big issue of your paper is that the results are a mix between results, methods, and discussion, moreover, the discussion section is missing. I made some important comments in the attached file to help you in improving your paper.

Round 2

Reviewer 1 Report

Como se mencionó anteriormente, el documento aborda un tema interesante; Enmiendas se han hecho de acuerdo con los comentarios de los revisores y el documento podría ser publicado después de correcciones menores (errores tipográficos).

Por favor, verifique:

Página 2 línea 54 ; Página 3 línea 96 y página 4 línea 163

Reviewer 2 Report

Dear authors, thankyou for agreeing with all my suggestions, you changed the paper in correct manner, I think now the manuscript is suitable for publication, best regrads